# Developing an implementation intervention, and identifying strategies for integrating health innovations in routine practice: A case study of the implementation of an insulin patient decision aid

**Wen Ting Tong** [1]*, **Yew Kong Lee**[1], **Chirk Jenn Ng**[1,2,3], **Ping Yein Lee**[4]

1 Department of Primary Care Medicine, Faculty of Medicine, Universiti Malaya, Kuala Lumpur, Malaysia,
2 Department of Research, SingHealth Polyclinics, Singapore, Singapore, 3 Health Services and Systems Research, Duke-NUS Medical School, Singapore, Singapore, 4 eHealth Unit, Faculty of Medicine, Universiti Malaya, Kuala Lumpur, Malaysia

* tongwenting@um.edu.my

## Abstract

### Background

Implementation, which is defined as the process of getting evidence-based innovation to be utilised is critical in ensuring innovation is being integrated into real-world practice. The way an implementation intervention (i.e., a bundle of strategies to facilitate implementation) is developed has an impact on the types of strategies chosen, and whether they are relevant to the implementation setting to exert their effects. However, literature pertaining to development of intervention or detailed descriptions of implementation processes are scarce. This study aims to report the development of an implementation intervention to integrate the use of an evidence-based innovation in routine practice, using a Malaysian insulin patient decision aid (PDA) as an exemplar.

### Methods

The development of the insulin PDA implementation intervention was divided into two phases, incorporating step 3 and 4 of the Action Cycle in the Knowledge to Action framework. In Phase 1, barriers to the insulin PDA implementation was explored through qualitative interviews using an interview guide developed based on the Theoretical Domains Framework. In Phase 2, prioritisation of the barriers was conducted using the multivoting technique. Next, potential strategies that can address the barriers were identified based on understanding the clinic context, and evidence from literature. Then, the selected strategies were operationalised by providing full descriptions in terms of its actor, action, action target, temporality, dose, implementation outcome affected, before they were embedded into the patient care pathway in the clinic. The implementation intervention was finalised through a clinic stakeholders meeting.

**Funding:** This study was supported by the University of Malaya Research Grant (UMRG) (RP041C-15HTM); and the University of Malaya Postgraduate Research Grant (PPP) (PG264-2016A). The funders had no role in study design, data collection and analysis, decision to publish, or preparation of the manuscript.

**Competing interests:** The authors have declared that no competing interests exist.

## Results

In Phase 1, a total of 15 focus group discussions and 37 in-depth individual interviews were conducted with: healthcare policymakers (n = 11), doctors (n = 22), diabetes educators (n = 8), staff nurses (n = 6), pharmacists (n = 6), and patients (n = 31). A total of 26 barriers and 11 facilitators emerged and they were categorised into HCP, patient, organisational, and innovation factors. The multivoting exercise resulted in the prioritisation of 13 barriers, and subsequently, a total of 11 strategies were identified to address those barriers. The strategies were mandate change, training workshop, involve patients' family members or caretakers, framing/reframing, inform healthcare providers on the advantages of the insulin PDA use, define roles and responsibilities, place the insulin PDA in the consultation room, provide feedback, systematic documentation, to engage patients in treatment discussions, and juxtapose PDA in preferred language with patient's PDA in their preferred language to help with translation.

## Conclusion

This study highlights main barriers to PDA implementation, and strategies that can be adopted for implementation. The steps for intervention development in this study can be compared with other intervention development methods to advance the field of implementation of evidence-based innovations.

## Introduction

Some health innovations fail to be translated into practice that can result in quality of care and meaningful patient outcomes despite being proven to be effective in efficacy studies [1]. Implementation, which is defined as the process of getting evidence-based innovation to be utilised in a setting is critical in ensuring innovation is being integrated into routine practice, and used correctly leading to the outcomes it is intended to achieve [2]. However, many barriers exist that hinder implementation and this includes healthcare providers (HCPs), patient, professional interactions, organisational structure, incentives and resources and social, political and legal factors [3].

One example of evidence-based innovation that is still underutilised despite its proven effectiveness is patient decision aids (PDA) [4]. PDAs are tools that help to facilitate shared decision making (SDM) between HCPs and patients to reach a treatment decision that is aligned with a patient's values and preferences [5]. PDA use can increase patient involvement in decision-making, reduce decisional conflict, and has a positive effect on patient-doctor communication [6], although its cost effectiveness from patient, organisation, and society perspective is still unclear [7]. Despite the promotion of PDA use through policies, regulations and laws, PDA implementation in routine practice is still hampered by various barriers such as a lack of resources, concerns about PDA quality and availability, HCPs' and patients' attitudes, and workflow processes [8].

Implementation strategies, which are defined as *"methods or techniques used to enhance the adoption, implementation, and sustainability of a clinical program or practice"* are key in ensuring that a health innovation can be effectively implemented [9]. The way an implementation intervention (i.e., a bundle of strategies to facilitate implementation) is developed has an

impact on the types of strategies chosen, whether they are relevant to the implementation setting to exert their effects and being adopted widely in the real world. However, current literature pertaining to development of implementation intervention or detailed descriptions of implementation processes are scarce, poorly reported or indexed [10]. Descriptions of conceptual frameworks underpinning the research, implementation context, and information about the implementation strategies are often lacking or missing [10–12]. Furthermore, the methods for tailoring strategies to contextual determinants is scarce [13].

The issues above extends to literature pertaining to PDA implementation, which mainly focuses on reporting barriers and facilitators to implementation [14–17]. How a PDA implementation intervention or plan is developed, and strategies that were adopted were often not reported or described in detail [18–20]. Among studies that did report, there is a lack of a systematic approach to PDA implementation intervention development [19, 21]. Furthermore, the use of implementation frameworks to guide PDA implementation is few. In some studies, the intervention development process was found to be too simplistic with some PDA implementers simply let implementation sites or HCPs decide how to implement the PDAs in their practice [19, 21, 22]. The lack of guidance and support has been reported to hamper effective implementation [23]. Furthermore, strategies that are effective for PDA implementation is not very well studied [4].

This paper aims to report the development of an implementation intervention to integrate the use of an innovation in routine clinical setting, using a Malaysian insulin PDA as an exemplar. In Malaysia, type 2 diabetes is a major threat to its population given its rising prevalence [24]. However, despite many patients with type 2 diabetes not achieving target glycaemic control (76.2%), only 21.4% were on insulin therapy [25]. In 2012 (with an update in 2014), an insulin PDA booklet was developed in Malaysia to promote patient centered care through fostering SDM practices and to help patients to make informed decisions about diabetes treatment [26, 27]. The insulin PDA was developed based on evidence from a systematic review, decision support theories and frameworks, and the needs of local healthcare policymakers, clinicians, and patients [26]. The booklet has 12 pages divided into five sections. The first and second sections provide information about diabetes and insulin, and the pros and cons of available treatment options. The third section entails the values clarification exercise where patients share what is important to them about insulin initiation and their concerns with insulin. Subsequently, patients are asked explicitly if they needed more support (section four) before finally making a decision (section five). The insulin PDA is available in the Malay, Chinese, Tamil, and English languages to cater to the Malaysian multiethnic and multilingual population. According to the International Patient Decision Aids Standards Collaboration, which assesses the quality and standards of a PDA, the insulin PDA scored 7/7 for decision aid criteria and 5/8 for the risk of bias criteria [28].

Transparent and explicit reporting of the processes that were undertaken to develop the implementation intervention can help to provide an example to other implementers. In addition, information pertaining to main potential barriers and facilitators to the insulin PDA implementation, and the strategies that could address the relevant barriers will also be highlighted as they are the outcomes from the intervention development process.

## Methods

### Implementation framework

The Knowledge to Action framework was selected as the process model to guide the implementation of the insulin PDA [29]. This framework was chosen because the framework provides a clear and broad practical step-by-step guidance on how to move evidence-based

innovation to practice. The framework also provides flexibility in terms of how each of the steps in the Action Cycle can be operated and can be combined with other theories or frameworks to enhance findings [30]. This framework had been used for implementation of PDAs [31].

The development of the insulin PDA implementation intervention was divided into two phases incorporating specific steps in the Action Cycle of the Knowledge to Action framework. Phase 1 was assessment of barriers to the insulin PDA implementation. This phase corresponds to step 3 of the Action Cycle 'Assess barriers to knowledge use'. Phase 2 corresponds to step 4 in the Action Cycle 'Select, tailor, implement intervention'. This phase involved selection and tailoring of strategies to overcome the barriers identified in phase 1. These two steps within the Action Cycle were used for the implementation intervention development as implementation is more likely to be effective in changing professional practice and healthcare outcomes if the strategies are tailored based on identified barriers and facilitators [32].

Usually, initial barrier or needs assessment would lead to many barriers, and there is a scarcity of details on how barriers were prioritised for the design of a tailored intervention [33, 34]. This current study explicitly employed a step on prioritisation of barriers, in which to the researcher's knowledge has been rarely reported in other intervention development or implementation studies. Therefore, step 4 of the Action Cycle was expanded to include the barrier prioritisation step in addition to select, tailor and implementation of the strategies. Fig 1 illustrates the theory of change and the intervention development steps of this study.

## Phase 1: Assessment of barriers and facilitators to implementation of the insulin PDA

This phase adopted the qualitative study design to explore perceived barriers This study design allowed exploration of the issues of implementation of health innovation, particularly on implementation of PDA in the Malaysian context. Within this phase, the Theoretical Domains Framework (TDF) [35] was chosen as the determinant framework to identify the barriers and facilitators to implement the insulin PDA. The framework was selected because behaviour change is a major contributing factor on the adoption of an innovation. In addition, the TDF is also a comprehensive framework as apart from covering an individual's rational and cognitive aspects, it also covers emotional as well as organisational and resource factors that could influence behavior change.

The Malaysian public healthcare setting was chosen as the study setting as majority of diabetes patients are managed in the public sector where healthcare services are subsidised by the government. Qualitative interviews were conducted at one academic primary care clinic in an urban government teaching hospital (the University of Malaya Medical Center (UMMC)) and five public community clinics. Healthcare policymakers, HCPs, and patients who fulfilled the study inclusion criteria were invited to participate in the study. The inclusion criteria for healthcare policymakers were those who were responsible for or involved in making decisions on whether a particular health intervention should be implemented in their healthcare settings. They set standards of care and implement programmes to improve diabetes care. These included the director-level hospital manager, endocrine and primary care medicine policymakers, hospital matrons, and clinic managers. HCPs included were doctors, diabetes educators and staff nurses who were involved in advising patients about insulin initiation. Patients with type 2 diabetes who have been advised to start insulin or were currently using insulin and those who had been seeking diabetes treatment in the clinic for more than one year were recruited. In order to achieve maximum variation, we recruited participants from different socio-demographic backgrounds (ethnicity, gender) and included those with and without

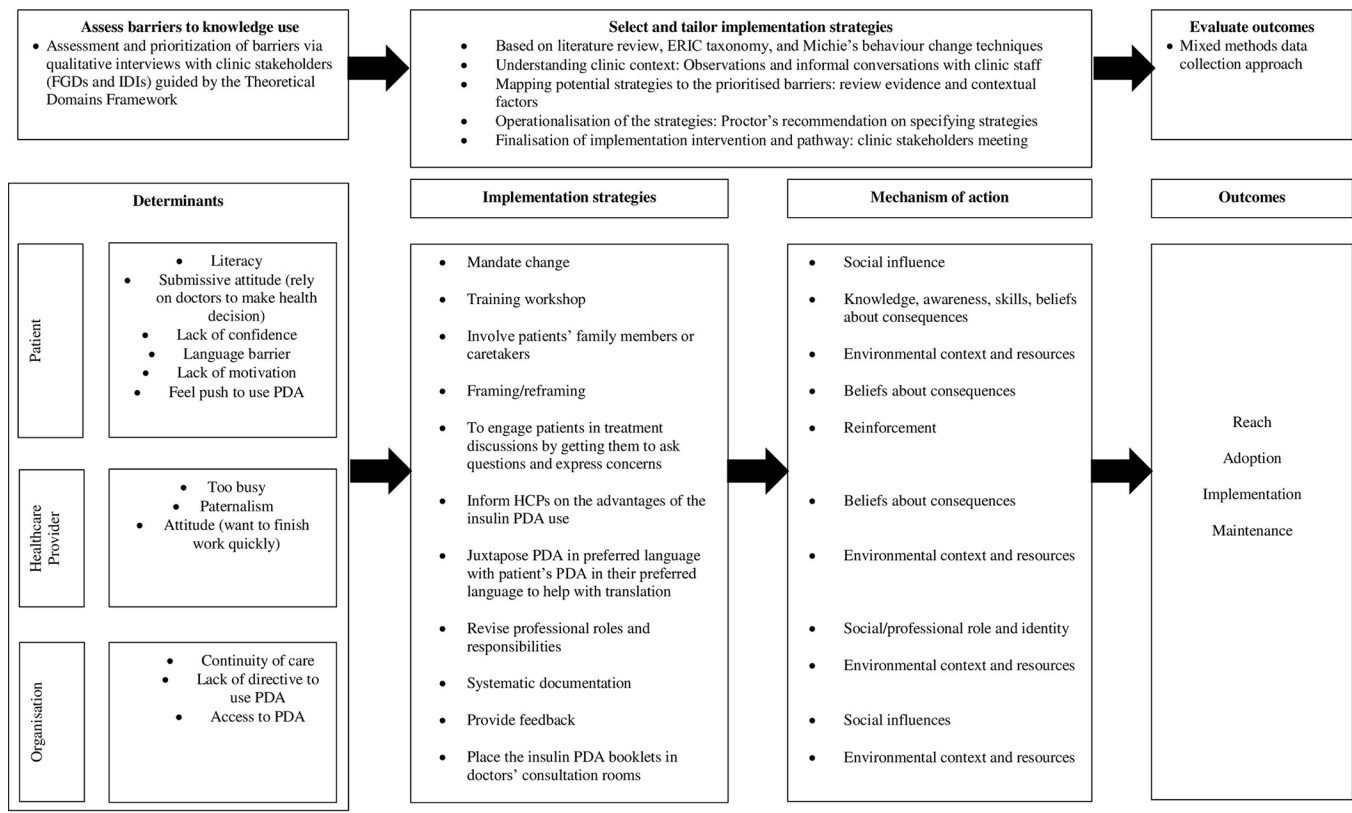

**Fig 1. Study theory of change and the implementation intervention development steps of this study.**

experience in using the PDA. Healthcare policymakers and HCPs who fulfilled the study inclusion criteria were invited to participate through telephone calls. Doctors in the clinics helped identify eligible patients and referred them to the researcher. Interview appointments were then made with eligible participants. Detailed information about the participant recruitment process have been reported in prior publications [36–38].

The interviews were conducted with the aid of a semi-structured interview guide which was developed based on the TDF [35], literature review on factors influencing PDA implementation, and discussions among researchers. The interview guide comprised of five sections, and they were: the innovation—insulin PDA, individuals, inner context–the clinic, and the outer context–the healthcare system. Questions and prompts were created according to the domains in the TDF within each of the sections. For example, under the Individual section, the question 'Do you think that insulin PDA will affect your/HCP's role? How?' was developed to address the social professional role and identity domain in the TDF. Under the inner context–clinic section, the question 'How do you think your organisation's working culture will affect the implementation of the PDA?' was developed to address the environmental context and resources domain in the TDF. The summary of the interview guide for policymakers and HCPs, and patients can be found in prior publications [36, 37]. The interview guides were adapted according to the participants' roles: healthcare policymakers, HCPs, and patients with type 2 diabetes.

Interviews were audio-recorded and transcribed verbatim. Thematic analysis was conducted using the NVivo qualitative analysis software. During the data analysis, the interview transcripts were read in detail and then coded. A code is short text that represents the meaning of a text segment. Codes that have similar meaning were grouped together to form a category,

and later the categories were compared and merged into bigger themes: HCP, patient, organizational, and innovation.

## Phase 2: Prioritisation of barriers, select, and tailor implementation intervention

**Setting selection for pilot implementation.** It has been recommended that implementation should begin small-scale by involving a limited number of motivated individual, teams or institutions in order for improvements of the intervention before expansion [39]. A pilot implementation of the insulin PDA was carried out at the UMMC primary care clinic. This clinic was chosen because it was where the insulin PDA was developed, and two of the researchers were affiliated with the clinic; CJN is a professor and family medicine specialist at the clinic while YKL is a senior lecturer at the Department of Primary Care Medicine, which is affiliated with the UMMC primary care clinic.

**Step 1: Prioritisation of barriers.** The multivoting technique [40] was employed to prioritise the barriers identified in Phase 1. Multivoting technique is a systematic process of a few voting rounds that helps to reduce a list of factors or barriers to a top few. The multivoting technique was chosen because it involves an objective and democratic process whereby every stakeholder in the clinic (stakeholders in this study included clinic managers, doctors, nurses and patients) can vote independently thus have an equal say in which barrier they think would hamper the insulin PDA implementation. Barriers, which are voted by all would be the ones that are important for all stakeholders and this is crucial, as every stakeholder will most likely have a direct role in the implementation of the insulin PDA.

The multivoting process involved the clinic stakeholders (n = 12) from the UMMC primary care clinic; the clinic coordinator, three doctors, two diabetes educators, three staff nurses and three patients in a two-round voting process, which took approximately three hours. During the multivoting exercise, the participants were informed that the purpose of the exercise was to narrow down the list of barriers to the insulin PDA implementation, that were identified in a previous qualitative study (Phase 1) so that most important barriers can be identified for development of targeted intervention. The researcher also presented the insulin PDA and the concept of SDM to the participants using PowerPoint slides. In addition, examples of how the insulin PDA can be used in the clinic were also provided to enable the participants to think about the processes of the insulin PDA implementation, and what are the main barriers that could hamper its implementation in the UMMC primary care clinic. Once the participants were familiar with SDM and the insulin PDA, a Voting Form was distributed which consisted of the barriers that were identified from Phase 1 (**S1 Table**). Facilitators identified in phase 1 were revised as barriers by expressing the idea in the form of negative statement. For example, 'Directive from higher authority' was revised to 'There is no clear directive from higher authority to use the insulin PDA'. This allowed the researcher to find out if the 'opposite' or absence of the facilitator would be a prioritised barrier for implementation.

The multivoting process is illustrated in Fig 2. Participants were given individual voting forms and asked to vote privately to prevent them from influencing one another. Once voting forms were completed, they were immediately entered into an excel spreadsheet and tabulated. The first round of voting resulted in 18 barriers remaining in the list. Then, participants were asked to vote for their top 10 highest priority items of the list. This reduced the number of barriers to approximately 10 items, which the researchers discussed and decided would be feasible for the development of the tailored intervention prior to the multivoting process.

**Step 2: Select and tailor strategies to overcome barriers to implementation.** Once the barriers were prioritised, the next step was selecting and tailoring strategies to overcome the

### Step 1: Round 1 voting

- Using the Voting Form developed (49 barriers), the participants voted as many items as they wish.

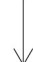

### Step 2: Round 1 analysis

- Once votes are tabulated, barriers with a vote count equivalent to or more than half the number of participants remain on the list.
- Second Voting Form generated.

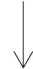

### Step 3: Round 2 voting

- Using the second Voting Form, (18 barriers) the participants voted for their top 10 highest priority items of the list

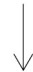

### Step 4: Round 2 analysis

- Round 1 analysis repeated.
- Key barriers identified.

**Fig 2. The multivoting process.**

barriers, and this involved three steps: 2a) searching for potential strategies, 2b) understanding the clinic context, and 2c) mapping strategies to prioritised barriers.

*2a) Searching for potential strategies.* A literature review was conducted to search for strategies that can overcome the prioritised barriers. In addition, strategies and processes proposed by participants in Phase 1 in overcoming specific barriers were another source for strategy identification and selection. The researcher (WTT) conducted a secondary analysis of all the interview transcripts in Phase 1. A total of 31 strategies were brought up by the participants (**S2 Table**). Lastly, lists of known types of strategies namely the Expert Recommendations for Implementing Change taxonomy [41], and the Michie's behaviour change technique (BCT) taxonomy [42] were also looked into. Relevant strategies found were mapped to the barriers that they can potentially address. This process was conducted by WTT.

*Step 2b*: *Understanding clinic context.* There was a need to understand implementation context to know which strategy would be feasible to be implemented, hence, WTT conducted observations and engaged in informal conversation with the clinic staff to understand the type 2 diabetes patient care pathway, and the availability of resources. The information gathered enabled the selection of strategies that would be appropriate and feasible to implement in the clinic, including incorporating them into the clinic's daily work processes without requiring too many changes. The key clinical contexts gathered for informing the insulin PDA implementation were:

- How were patients offered insulin therapy? (e.g.: where, by who, when)

- What happens to patients when a decision about insulin is made?

- What were the specific roles played by the various HCPs in the clinic?

- What were the available resources in the clinics (i.e.: infrastructure, staff)?

*Step 2c)*: *Mapping potential strategies to prioritised barriers*. Using all the information gathered from the literature and understanding of the clinic context, the researchers then decided on the potential strategies to be utilised to overcome the prioritised barriers. The researcher went through each of the strategies identified and assessed its appropriateness based on evidence of its effectiveness and feasibility in implementing in the UMMC primary care clinic.

**Step 3: Operationalisation of the strategies.** Subsequently, each of the discrete strategy that was selected was operationalised. The reason for specifying the strategies was to allow the researcher to determine if the strategies would be feasible to be carried out in the clinic context. Also, when a strategy is described in sufficient detail, it allows for scaling or replication in other research and practice [43]. The operationalisation was based on recommendations by Proctor et al., (2013) who suggested that implementation strategies should be specified according to seven dimensions: the actor, the action, the action target, temporality, dose, the implementation outcome affected and, the justification for selecting the strategy [43]. However, in this study, justification for selection of the strategy had already been described in step 2c, thus, was not included as part of the operationalisation. The operationalisation of the strategies was done by the researchers who took into account the clinic's context and resources. One of the researchers (CJN) is a family medicine specialist practising in the clinic and was able to provide input on the workflow processes in the UMMC primary are clinic. Once the implementation strategies were operationalised, they were embedded into the existing diabetes patient care pathway in the clinic.

**Step 4: Finalisation of implementation intervention and pathway through a clinic stakeholders meeting.** A clinic stakeholders meeting was conducted to discuss and finalise the implementation intervention. Clinic stakeholders who participated in the multivoting exercise in Step 1 were invited. The participants were engaged in an open discussion to discuss and come to a consensus on the implementation intervention drafted by the researchers. The proposed strategies to overcome the prioritised barriers, and the PDA implementation flow diagram were presented and explained in detail. Then, participants were asked to take their time to deliberate if the draft intervention would be effective and feasible to be implemented in the clinic setting. Every stakeholder (doctors, diabetes educators and staff nurses) was also asked to peruse their implementation tasks.

The open group discussion enabled the participants of various stakeholders to counter and support each other's opinions on the draft implementation plan. This led to an agreement on which strategies to be retained or dropped. The meeting lasted for one and a half hours. Once consensus was reached, an implementation date was set. The outcomes of the implementation of the insulin PDA in this study was reported in another paper.

## Ethics considerations

This study received ethics approval from the Medical Research and Ethics Committee Ministry of Health Malaysia (NMRR-15-1598-27260) and the University of Malaya Medical Centre Medical Ethics Committee (reference: MECID.NO: 20158–1600). Informed verbal and written voluntary consent were sought from all participants and all data were treated with confidentiality. The data collection was carried from 1 Dec 2015 to 30 September 2016.

## Results

### Phase 1: Assessment of perceived barriers to the insulin PDA implementation

A total of 15 focus group discussions and 37 in-depth individual interviews were conducted with: healthcare policymakers (n = 11), doctors (n = 22), diabetes educators (n = 8), staff

nurses (n = 6), pharmacists (n = 6), and patients (n = 31). Among the healthcare policymakers and the HCPs, their mean age was 39.4 (± standard deviation (SD) 9.7) years (range: 28–57) and 86.8% were female. The mean duration of practice since entering workforce was 14.7 (± SD 10) years (range: 3–35). Six of them had experience using the insulin PDA from the acceptability study. Among the 31 patients, their mean age was 63.0 (± SD 11.6) years (range: 36–79) and there was slightly more male participants (51.6%). The mean duration of having diabetes was 12.8 (± SD 7.9) years (range: 4 months– 30 years) while the mean duration of them seeking diabetes treatment in the clinic was 8.58 (± SD 5.9) years (range: 2 months– 20 years). Almost half of them were Malays (48.4%) followed by Chinese (32.3%) and Indians (19.3%). Slightly more than half of the patients (54.8%, n = 17/31) were using insulin at the time of the interview.

The assessment of barriers and facilitators to the insulin PDA implementation at the six clinics uncovered a total of 26 barriers and 11 facilitators categorised into HCP, patient, organisational, and innovation factors (Table 1).

## Phase 2: Prioritization of barriers, select, and tailor implementation intervention

**Prioritisation of barriers.**    The multivoting exercise resulted in the prioritisation of 13 barriers from the initial list of barriers. The 13 prioritised barriers are indicated with an asterisk in Table 1.

**Select and tailor strategies to overcome barriers to implementation.**    Table 2 shows the potential strategies from various sources that could address the prioritised barriers. Based on findings from the literature review, and consideration each strategy's effectiveness and feasibility based on clinic context, relevant strategies were selected by the researchers. The strategies selected are bolded in Table 2. Some strategies selected could address multiple barriers. For example, the strategy 'Framing and reframing' was selected to address the barriers 'HCPs will not use the PDA in the clinic because they are too busy as there are too many patients', and 'Patients feel that the insulin PDA is a tool to persuade them to start insulin'. In addition, some strategies are similar across the various sources. For example, placement of PDAs in consultation rooms can be found under the sources 'Effective strategies from literature on PDA implementation', and 'Strategies proposed by participants in Phase 1 study'. Similar strategies are also bolded in Table 2.

Similar strategies were consolidated. Finally, a total of 11 strategies were selected by the researchers based on evidence in the literature, the feasibility to implement in the clinic context, and their expected mechanism of action. The strategies are listed in Table 3.

For example, the strategy "mandate change" was chosen to address the barrier "not having a clear directive from the top management to use the insulin PDA". This strategy was selected because it has been shown that when organisational leader showed clinicians that SDM was an important organisational priority, it led clinicians to believe that SDM was part of the organisation's work practice and not something that was imposed on them [46, 64, 65] (**evidence**). Based on the clinic context, the Head of Department or the clinic coordinator organise monthly unit meetings, where all clinic staff (doctors, diabetes educators, staff nurses, appointment clerks) gathers to discuss matters related to the clinic, Hence, this provides opportunity to implement this strategy in a unit meeting (**clinic context**). Participants from Phase 1 also noted that directive from higher authority was an effective way to implement the insulin PDA. Furthermore, this strategy was perceived to be more feasible compared to other strategies such as incorporating the insulin PDA use as part of key performance index or standard operating procedure. This is because the latter strategies would require approvals from hospital authorities and bureaucratic paperwork that are time consuming (**clinic context**). Declaration from

**Table 1. Perceived barriers and facilitators to implementation of the insulin PDA.**

| Category | Theme |
|---|---|
| **Organisational** | |
| B: Time constraint due to high patient and work load | 1. B: HCPs are too busy as there are too many patients* |
| B: Lack of manpower | |
| B: Cannot access the insulin PDA | 2. B: Lack of accessibility to the insulin PDA* |
| B: Lack of awareness on where to get the insulin PDA | |
| B: Lack of funding to print the insulin PDA booklets | 3. B: Lack of funding to print the insulin PDA booklets |
| B: Lack of continuity care | 4. B: HCPs will not use the PDA in the clinic because they will not be able to see the same patient to follow up on the PDA* |
| B: Uncertain of the insulin PDA implementation process | |
| B: Lack of teamwork | 5. B: Lack of teamwork |
| B: Competing programmes in the clinic | 6. B: Competing programmes in the clinic |
| F: Directive from higher authority | 7. F: Directive from higher authority (Revised as barrier statement: HCPs will not use the PDA in the clinic because there is no clear directive to use the PDA)* |
| F: Having a senior role model | 8. F: Role model and staff motivation |
| F: Acknowledgement on using the insulin PDA by clinic leader | |
| F: Having a dedicated staff or team to carry out the implementation | 9. F: Having a dedicated staff or team to carry out the implementation |
| B: Lack of rooms in the clinic for insulin PDA discussions | B: Lack of rooms in the clinic for insulin PDA discussions |
| **HCP** | |
| B: Perception that insulin PDA use is a doctor's task | 10. B: HCPs will not use the PDA in the clinic because they are not in-charge to use of the PDA in the clinic (Role boundary)* |
| B: Perception that using the insulin PDA is not within their job scope | |
| B: Lack of awareness of the insulin PDA | 11. B: Lack of awareness of the insulin PDA |
| B: Perception that insulin PDA use is an added workload | 12. B: Perception that insulin PDA use is an added workload |
| B: Lack of familiarity with the insulin PDA | 13. B: Lack of familiarity with the insulin PDA |
| B: HCPs want to finish their work quickly | 14. B: HCPs want to finish their work quickly* |
| B: HCPs lack of shared decision making approach | 15. B: HCPs tend to make decisions for their patients instead of practising SDM* |
| B: HCPs paternalistic attitude i.e. make health decisions for patients | |
| B: Doctors do not like patients to ask many questions | |
| B: Peer influence | 16. B: Peer influence |
| B: HCPs are not clear when delivering information to patients | 17. B: HCP's poor communication skills |
| B: HCPs talk to fast to patients | |
| F: HCPs talk in simple and laymen terms to patients | |
| F: HCPs belief that the insulin PDA is useful | 18. F: Belief that PDA is useful and beneficial |
| F: HCPs belief that insulin PDA is beneficial | |
| F: Trust in the credibility of the insulin PDA | 19. F: Trust in the PDA credibility |
| F: HCPs curiosity on the effectiveness of the innovation | 20. F: Motivation to try new innovation (Revised as barrier statement: HCPs will not use the PDA in the clinic because they are not motivated to try new innovations*) |
| F: HCPs desire to improve on personal advancement and patient care | |

*(Continued)*

**Table 1.** (Continued)

| Category | Theme |
|---|---|
| **Organisational** | |
| F: Having reminders | 21. F: Having reminders |
| F: Awareness of the positive outcomes of the insulin PDA | 22. F: Awareness of the positive outcomes of the insulin PDA |
| **Patient** | |
| B: Patients cannot understand the insulin PDA | 23. B: Patients cannot read or understand the insulin PDA* |
| B: Patients' health condition (e.g: poor vision) render difficulty in using the insulin PDA | |
| B: Patients are not willing to pay for the insulin PDA | 24. B: Patients are not willing to pay for the insulin PDA |
| B: Patients let doctors or relative to make health decisions for them | 25. B: Patient rely on doctors to make health decisions*<br>B: Patient are not confident to use the PDA by themselves* |
| B: Patients lack of confidence in using the insulin PDA by themselves to make decision | |
| 26. B: Patients lack of motivation and effort to improve health | B: Patients lack of motivation and effort to improve health |
| B: Patients do not like to read | 27. B: Patient do not like to read |
| B: Patients may feel push to use insulin when receiving the insulin PDA | 28. B: Patient feel that the insulin PDA is a tool to persuade them to start insulin* |
| B: Patients are busy | 29. B: Patients are busy |
| B: Language barrier between HCP and patient | 30. B: Language barrier between HCP and patient* |
| B: Difficulty to return to the clinic for PDA follow-up | 31. B: Difficulty to return to the clinic for PDA follow-up |
| F: Patients' trust in doctor | 32. F: Patients' trust in doctor |
| F: Belief that PDA is beneficial | 33. F: Belief that insulin PDA is beneficial |
| **Innovation** | |
| B: Insulin PDA has too many pages and wordy | 34. B: Insulin PDA has poor design |
| B: Insulin PDA is not visually attractive | |
| B: Insulin PDA may delay treatment in patient | 35. B: Perception of negative consequences from the insulin PDA use |
| B: Insulin PDA may replace consultation | |
| B: Disagree with the content of the insulin PDA | |
| B: Insulin PDA may caused patients to feel confuse | |
| B: Insulin PDA may caused patients to feel anxious | |
| F: Insulin PDA comprehensive and informative | 36. F: Insulin PDA comprehensive and informative |

B: Barrier; F: Facilitator

*Prioritised barrier

the clinic authority may influence the clinic staff to see that the insulin PDA implementation is a clinic's priority and will influence them to be involved in the implementation. The expected mechanism of action for this strategy is social influence **(mechanism of action).** The justification for all the strategies selected, their expected mechanisms of action, and how they can address the barriers can be found in **S3 Table.**

**Strategies operationalized.** For each of the strategy selected, full descriptions were provided in terms of actor, action, action target, temporality, dose, implementation outcome affected. Table 4 shows an example of the operationalisation of the strategy mandate change to overcome the prioritised barrier "HCPs will not use the PDA in the clinic because there is no clear directive from the top management to use the PDA". The operationalisation for other strategies can be found in **S4 Table**.

**Table 2. Strategies selected to address specific barriers in the insulin PDA implementation from various sources.**

| | Barrier | Effective strategies from literature on PDA implementation | Strategies proposed by participants in Phase 1 study | Strategies recommended in the literature on PDA implementation | Strategies recommended in other literature (not PDA implementation literature) to address specific barriers | Strategies identified in the ERIC taxonomy and Michie BCCTv1 taxonomy |
|---|---|---|---|---|---|---|
| 1. | HCPs will not use the PDA in the clinic because they are too busy as there are too many patients | • **Conduct training to increase healthcare professionals' knowledge and skills in using patient decision aids** [22, 31, 44–47]<br><br>• Giving PDA to patients pre-visit [20, 48, 49]<br><br>• Using IT system to delivery PDA to patients (e.g: EMR) [50, 51]<br>• Involvement of staff other than doctors in the implementation [49, 52] | • **Conduct SDM and insulin PDA training**<br><br>• To involve diabetes educators or nurses to use insulin PDA with patients<br><br>• HCP to focus only on important topics when using the insulin PDA<br><br>• To let patients read the insulin PDA at their own time and discuss in the next visit<br><br>• To get patients to use PDA prior to consultation<br><br>• To have a person in charge or a dedicated staff or team for the insulin PDA implementation | • Demonstration on how PDA can be used during consultations [53]<br><br>• Inform HCPs that PDA does not necessarily increase consultation time and can help to save time in the future (Framing and reframing) [54]<br><br>• Remind clinicians [55, 56]<br><br>• Patient self-access to PDAs [44, 57] | | • **Conduct educational meetings**<br><br>• BCT: Demonstration of the behavior (_ ((((xxx[41, 42]<br><br>• **Revise professional roles**<br><br>• **BCT: Framing/ reframing** |
| 2. | HCPs will not use the PDA in the clinic because they will not be able to see the same patient to follow up on the PDA | | • **Systematic documentation** | | | • Prompts/cues |
| 3. | HCPs tend to make decisions for their patients instead of practising SDM | • **Conduct training to increase healthcare professionals' knowledge and skills in using patient decision aids** [22, 31, 44–47] | • **Conduct SDM and insulin PDA training** | | | • BCT: Information about social and environmental consequences; BCT: Information about emotional consequences **(Inform HCPs on the advantages of the insulin PDA use)** |

(*Continued*)

**Table 2.** (Continued)

| | Barrier | Effective strategies from literature on PDA implementation | Strategies proposed by participants in Phase 1 study | Strategies recommended in the literature on PDA implementation | Strategies recommended in other literature (not PDA implementation literature) to address specific barriers | Strategies identified in the ERIC taxonomy and Michie BCCTv1 taxonomy |
|---|---|---|---|---|---|---|
| 4. | Patients cannot read or understand the insulin PDA | | • **Involve patients' family members or caretakers**<br><br>• **To involve diabetes educators or nurses to use insulin PDA with patients**<br><br>• HCPs to go through the insulin PDA with patient<br><br>• To identify patients who are eligible to use the insulin PDA | • Having multiple reinforcing contacts to support active decision-making in the delivery of the PDAs [58]<br><br>• Demonstrating and modeling values clarification and physician interactions in PDAs [58]<br><br>• Delivery of a PDA by a health professional [58] | • Literacy appropriate communication strategies [59]<br><br>• Use of tape recorders, writing information down for patients [59] | **Revise professional roles** [**41**, **56**, **59**, **60**] |
| 5. | Patients rely on doctors to make health decisions | • **Increase patient engagement in health decision making by enabling patients to feel comfortable asking questions and expressing their preferences** [47, 61] | • To identify patient who are eligible to use the insulin PDA | | • **Patients should be encouraged to express their values and ask questions** [59]<br><br>• Identify a systematic and efficient approach for encouraging patients to play a more active role. [59] | • Prepare patients/consumers to be active participants [60]<br><br>• BCT: Verbal persuasion about capabilities (_ ((((xxx[42]<br><br>• BCT: Information about social and environmental consequences; BCT: Information about emotional consequences (Inform HCPs on the advantages of the insulin PDA use) |
| 6. | Patients feel that the insulin PDA is a tool to persuade them to start insulin | | • Get HCPs to communicate to patient about the insulin PDA in a non-threatening manner | | | • BCT: Information about social and environmental consequences; BCT: Information about emotional consequences (Inform HCPs on the advantages of the insulin PDA use)<br><br>• **BCT: Framing and reframing** (_ ((((xxx[**42**] |

(*Continued*)

**Table 2.** (*Continued*)

| | Barrier | Effective strategies from literature on PDA implementation | Strategies proposed by participants in Phase 1 study | Strategies recommended in the literature on PDA implementation | Strategies recommended in other literature (not PDA implementation literature) to address specific barriers | Strategies identified in the ERIC taxonomy and Michie BCCTv1 taxonomy |
|---|---|---|---|---|---|---|
| 7. | HCPs will not use the PDA in the clinic because they are not in-charge to use of the PDA in the clinic | • **Assign dedicated staff for implementation** [51, 62] | • To have a person in charge or a dedicated staff or team for the insulin PDA implementation | | | • **Revise professional roles** [41, 56, 59, 60] |
| 8. | Patients are not confident to use the insulin PDA by themselves | • **Increase patient engagement in health decision making by enabling patients to feel comfortable asking questions and expressing their preferences** [47, 61] | • To get HCPs to introduce the insulin PDA to patients<br><br>• To involve diabetes educators or nurses to use the insulin PDA with patients<br><br>• To identify patients who are eligible to use the insulin PDA | | | • Prepare patients/consumers to be active participants [41, 56, 59, 60]<br><br>• BCT: Verbal persuasion about capabilities (_ ((((xxx[42]<br><br>• BCT: Information about social and environmental consequences; BCT: Information about emotional consequences (Inform HCPs on the advantages of the insulin PDA use)<br><br>• **Revise professional roles** |
| 9. | Patients have difficulty to use the insulin PDA with HCPs because of language barrier | | • **Juxtapose PDA in preferred language with patient's PDA in their preferred language to help with translation**<br><br>• Refer patients to HCPs who can speak the same language as the patients<br><br>• Involve patients' family members or caretakers<br><br>• HCPs to go through the insulin PDA with patient | | • Use of trained interpreters [59]<br><br>• Increase bilingual HCPs [63]<br><br>• Use professional interpreting services [63]<br><br>• Use ad hoc interpreters (family members, friends, untrained medical, non-medical staff, and strangers) [63] | |

(*Continued*)

**Table 2.** (Continued)

| | Barrier | Effective strategies from literature on PDA implementation | Strategies proposed by participants in Phase 1 study | Strategies recommended in the literature on PDA implementation | Strategies recommended in other literature (not PDA implementation literature) to address specific barriers | Strategies identified in the ERIC taxonomy and Michie BCCTv1 taxonomy |
|---|---|---|---|---|---|---|
| 10. | HCPs will not use the PDA in the clinic because they don't know where to get the PDA | • **Physicians reported to distribute PDAs to patients more when they are placed in the consultation rooms as compared to ordering one from medical assistants** [52] | • Conduct SDM and insulin PDA training<br><br>• **Place the insulin PDA booklets in doctors' consultation rooms**<br><br>• Develop the insulin PDA implementation pathway | | | |
| 11. | HCPs will not use the PDA in the clinic because there is no clear directive to use the PDA | • **Mandate and organisational leadership** [46, 64, 65] | • Issue directive to implement the insulin PDA<br>• Incorporate insulin PDA use as part of key performance index, standard operating procedures or in the diabetes management guideline | | | • **Mandate change** |
| 12. | I think HCPs will not use the PDA in the clinic because they want to finish their work quickly | | | | | • **BCT: Information about social and environmental** consequences; BCT: Information about emotional consequences (**Inform HCPs on the advantages of the insulin PDA use**)<br><br>• BCT: Demonstration of the behavior |
| 13. | HCPs will not use the PDA in the clinic because they are not motivated to try new innovations | • Conduct training to increase healthcare professionals' knowledge and skills in using PDAs [22, 31, 44–47]<br><br>• **Provide feedback** [64–69]<br><br>• Identify and prepare champion [21, 49, 66, 69]<br><br>• Provide incentives (e.g.: money incentives, CME credits) [44, 49, 55, 67]<br><br>• PDA implementation is facilitated by HCPs' perception that SDM and PDA will have a positive impact on the health care processes and patient outcomes [70]. | • Inform HCPs on the advantages of the insulin PDA use<br><br>• Conduct SDM and insulin PDA training<br><br>• **Monitoring and provide feedback**<br><br>• Have a champion<br><br>• Provide incentive or reward | | | • **BCT: Information about social and environmental** consequences; BCT: Information about emotional consequences (**Inform HCPs on the advantages of the insulin PDA use**)<br><br>• **Audit and provide feedback** |

**Table 3. Implementation strategy selected and their expected mechanisms of action.**

| Strategy | Expected mechanism of action |
|---|---|
| 1. Mandate change | Social influence |
| 2. Training workshop | Knowledge, awareness, skills, beliefs about consequences |
| 3. Involve patients' family members or caretakers | Environmental context and resources |
| 4. Framing/reframing | Beliefs about consequences |
| 5. To engage patients in treatment discussions by getting them to ask questions and express concerns | Reinforcement |
| 6. Inform HCPs on the advantages of the insulin PDA use | Beliefs about consequences |
| 7. Juxtapose PDA in preferred language with patient's PDA in their preferred language to help with translation | Environmental context and resources |
| 8. Revise professional roles | Social/professional role and identity |
| 9. Systematic documentation | Environmental context and resources |
| 10. Provide feedback | Social influences |
| 11. Place the insulin PDA booklets in doctors' consultation rooms | Environmental context and resources |

Once the implementation strategies were operationalised, they were embedded into the existing diabetes patient care pathway in the clinic. This led to development of the draft implementation intervention (**S1 Fig**). The process of pre-visit PDA use prior to consultation was included as literature has shown that pre-visit PDA use by patients led to more effective use of consultation time as doctors focused on discussions about treatment choices rather than communicating facts and information [20, 48, 49]. This would involve diabetes educators identifying patients who are eligible to receive the insulin PDA using the electronic medical record and giving the insulin PDA to patients. Patients can choose to read by themselves or ask diabetes educator or staff nurse to help them to go through the insulin PDA with them prior to their consultation with doctors. The process of letting patients take the insulin PDA home was also included as patients would have more time to go through the information and they can discuss with their family members. Patients would be given an appointment of within 3 months for follow-up on the insulin PDA use.

**Finalisation of implementation intervention and pathway through a clinic stakeholders meeting.** The clinic stakeholders meeting involved two clinic coordinators, one nursing officer, two doctors, one diabetes educator, four nurses, and two patients (n = 12). Nine participants were from the previous multivoting process. During the meeting, the strategies and their targeted barriers were explicitly described. The implementation intervention or plan was described, and the roles and tasks of each stakeholder in the clinic were explained.

**Table 4. Operationalisation of the strategy mandate change.**

| | | Dimensions to specifying a strategy | | | | | |
|---|---|---|---|---|---|---|---|
| Strategy | Target barrier | Actor | Action | Action target | Temporality | Dose | Implementation outcomes likely affected |
| 1. Mandate change | I think HCPs will not use the PDA in the clinic because there is no clear directive from the top management to use the PDA | Head of department or clinic coordinator | Declare to all staff about clinic's intention to implement and support the practice of SDM through the use of insulin PDA during unit meeting | Social influence: All clinic staff (doctors, diabetes educators, staff nurses, appointment clerks) would be influenced by the clinic authority to implement the insulin PDA | At the nearest unit meeting towards the insulin PDA training workshop | Two times per year following doctor turnover in clinic | Reach (doctor) Adoption (doctor) |

One prominent issue that was raised by the participants on the draft intervention was that diabetes educators would no longer be available in the clinic as they would be transferred to another unit. Hence, the strategies and processes that would involve diabetes educators to execute the pre-visit insulin PDA use, and referral of patients to diabetes educators for detail use of the insulin PDA would not be feasible. It was decided that the strategy 'Define roles and responsibilities' would replace the strategy 'Revise professional roles' where all the clinic staff (doctor, staff nurses, and appointment clerks) would be informed of their specific task in the insulin PDA implementation. In this strategy, all the clinic staff (doctor, staff nurses, and appointment clerks) would be informed of their specific task in the insulin PDA implementation (**action**). This strategy aimed to make HCPs embrace the implementation tasks given to them as part of their professional roles (**action target**). Clear role definitions have been reported to facilitate PDA implementation [23, 66] (**evidence**) by enabling clinic staff to be aware of one's own responsibility but at the same time also aware of other's unique contribution to the implementation and work together to accomplish tasks through shared responsibility [66]. Doctors and nurses would be informed explicitly of their specific implementation tasks during the training workshop. The flow of the insulin PDA implementation will be explained so that everyone would be aware of their responsibilities. Apart from doctors and nurses, appointment clerks would also be informed of their administrative role of to indicate that the insulin PDA had been given to patients in the patient medical record. The assignment of implementation tasks to various HCPs would be done one time (**dose**) prior to the implementation (**temporality**). This strategy aimed to make HCPs embrace the implementation tasks given to them as part of their professional roles. The expected mechanism of action for this strategy is social/professional role and identity.

Table 5 shows the finalised strategies selected to overcome specific barriers, while Fig 3 shows the finalised implementation intervention.

## Discussion

### Overview of findings

In this study, the development process of an implementation intervention was described: 1) conducting qualitative interviews with clinic stakeholders and patients to explore barriers and facilitators, 2) prioritisation of barriers using the multivoting technique, 3) searching for potential strategies that can address the barriers, 4) understanding clinic context, 5) mapping potential strategies to priorised barriers, 6) operationalisation of strategies by the researcher. and 7) finalisation of the intervention through a clinic stakeholders meeting. The result was an implementation intervention, which comprised 11 strategies selected to address 13 barriers based on the clinic context. This study also highlights important barriers to implementing PDA such as time constraint [71, 72], patient literacy level [49, 67, 73] and, access to PDAs [19, 55]. These barriers have been reported in other studies and therefore the strategies developed in this study may be applicable to other settings.

The strategies identified in this study such as making PDA accessible, training on the use of PDA, embedding PDAs within the care pathway, and allied HCPs engagement in implementation were also identified as key strategies for PDA implementation in another study [74]. However, in this study, the strategy of involving diabetes nurses to execute PDA delivery to patients was unable to be conducted due to the lack of manpower. This barrier is a deterrent to the formation of a multidisciplinary team in PDA implementation, which is crucial for effective implementation of PDAs [75]. When team involvement is not feasible, this indicates that the PDA implementation would have to rely on physicians to distribute PDAs. Hence strategies that target the physicians need to be enhanced. One way is incentivising PDA use, by motivating HCPs to be involved in

**Table 5. Strategies selected to overcome barriers to the insulin PDA implementation.**

| Strategy Barriers | 1 Manda-te change | 2 Training workshop | 3 Involve patients' family members or careta-kers | 4 Framing/ reframing | 5 To engage patients in treatment discussions by getting them to ask questions and express concerns | 6 Inform HCPs on the advanta-ges of the insulin PDA use | 7 Juxtapose PDA in preferred language with patient's PDA in their preferred language to help with translation | 8 Define roles and responsibilities | 9 Systema-tic documentation | 10 Provide feedback | 11 Place the insulin PDA booklets in doctors' consulta-tion rooms |
|---|---|---|---|---|---|---|---|---|---|---|---|
| 1. HCPs are too busy as there are too many patients | | x | | x | | | | | | | |
| 2. HCPs tend to make decisions for their patients | | x | | | | x | | | | | |
| 3. HCPs want to finish their work quickly | | | | | | x | | | | | |
| 4. HCPs are not motivated to try new innovations | | | | | | x | | | | x | |
| 5. Patients cannot read or understand the insulin PDA | | | x | | | | | | | | |
| 6. Patients rely on doctors to make health decisions | | | | | x | | | | | | |
| 7. Patients feel that the insulin PDA is a tool to persuade them to start insulin | | | | x | | | | | | | |
| 8. Patients are not confident to use the insulin PDA by themselves | | | | | x | | | | | | |
| 9. Language barrier | | | | | | | x | | | | |

*(Continued)*

**Table 5.** (Continued)

| Strategy Barriers | 1 Manda-te change | 2 Training workshop | 3 Involve patients' family members or careta-kers | 4 Framing/ reframing | 5 To engage patients in treatment discussions by getting them to ask questions and express concerns | 6 Inform HCPs on the advanta-ges of the insulin PDA use | 7 Juxtapose PDA in preferred language with patient's PDA in their preferred language to help with translation | 8 Define roles and responsibilities | 9 Systema-tic documentation | 10 Provide feedback | 11 Place the insulin PDA booklets in doctors' consulta-tion rooms |
|---|---|---|---|---|---|---|---|---|---|---|---|
| 10. HCPs not able to see the same patient to follow up on the PDA | | | | | | | | | x | | |
| 11. HCPs will not use the PDA in the clinic because they are not -in-charge to use of the PDA in the clinic | | | | | | | | x | | | |
| 12. HCPs will not use the PDA in the clinic because they don't know where to get the PDA | | | | | | | | | | | x |
| 13. HCPs will not use the PDA in the clinic because there is no clear directive to use the PDA | x | | | | | | | | | | |

the implementation, or, specifically making PDA use as an alternative means of informed consent to protect HCPs from 'failure to inform' lawsuit [76]. At the broader level, there is a need to inculcate SDM culture and PDA use among HCPs. HCPs need to understand their role in SDM in terms of providing quality information to patients and supporting patients in their deliberation of the treatment option. Hence, HCP training is warranted.

## Consideration of the steps involved in the intervention development process

There are many steps that can be taken when developing implementation interventions. This study followed the steps entailed in the Knowledge to Action Framework's action cycle. The

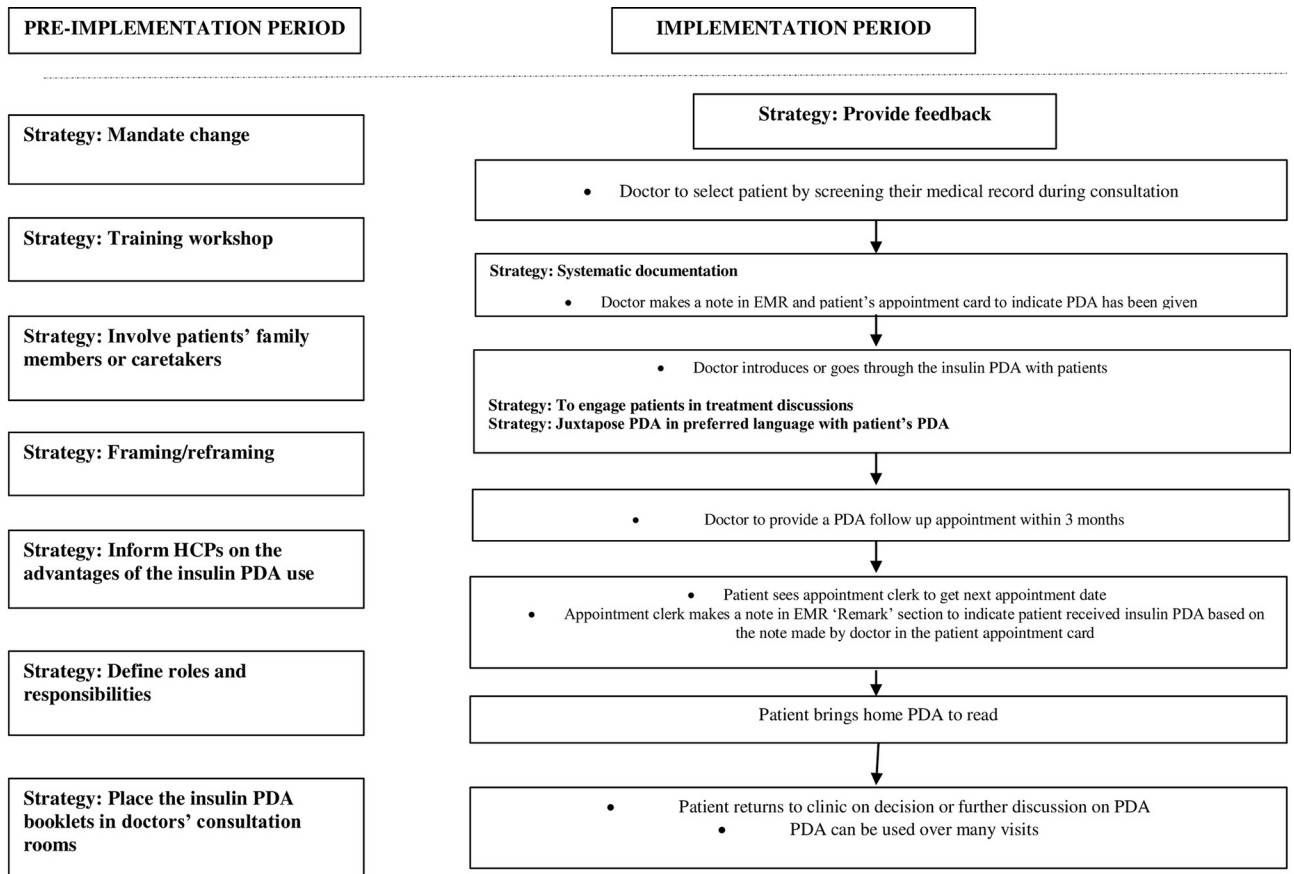

**Fig 3. The finalized implementation intervention.**

steps taken in this study, namely barrier identification, mapping of strategies to barriers, and engaging stakeholders were also commonly found in many other intervention development efforts. These steps can be considered as fundamental when designing an implementation intervention [77]. It is important to consider the steps to be undertaken for intervention development as each additional step adds to the complexity of the process of intervention development, and incurs burden to stakeholders and the resources and time needed for implementers [78–80]. Powell et al., (2017) proposed the use of concept mapping, group model building, conjoint analysis, and intervention mapping for intervention development [13, 81]. While these methods are highly participatory and provide concrete steps for facilitating selection and tailoring of strategies, however, these methods also required training and methodological consultation as methods such as group model building and concept mapping involves methematical modelling and use of proprietary software [13]. The challenges of using conjoint analysis and the intervention mapping approach have been demonstrated in some studies where modifications of the approaches were needed to lower costs, increase accessibility of the participation of the stakeholders in the process [78], and reduce complexity for intervention development [82].

Albeit various steps to develop an intervention were reported in many studies, it is hard to tell which step or process (a combination of steps and their sequence) is more superior to others. Each of these step or processes has its own merit in terms of leading to a more robust intervention being developed. Nevertheless, in this current study, based on the researchers' experience, the entire development process of the insulin PDA intervention was relatively low

cost, simple, pragmatic, and not overly burdensome to stakeholders; hence making it potentially replicable in other resource-constrained settings.

## Simplifying the barrier prioritisation process

This study adopted an implementation step that was rarely noted in implementation process models namely prioritisation of barriers. This step is crucial as addressing too many barriers would not be feasible given the limited time and resources, while at the same time, identifying the more important barriers to address would lead to a higher chance for effective implementation. In this study, a relatively easy and simple approach namely the multivoting technique was selected to prioritise the list of barriers identified in Phase 1. Furthermore, the use of the multivoting approach provided a systematic and democratic process of reaching key barriers that need to be overcome compared to methods adopted in other studies whereby barriers were prioritised based on the ones that were most commonly brought up during FGDs [83], and discussions among the research team [84].

Prioritisation may be based on various priority criteria such as the impact of problem, size of the problem, urgency of solving the problem, availability of solutions, availability of resources, and, cost and/or return on investment [85]. However, in this study, the participants were asked to vote for barriers based on the degree of which the barrier would most likely hamper the insulin PDA implementation. Other studies have reported to use more than one prioritisation criteria. In a study to prioritise barriers associated with implementation of clinical care elements for acute stroke management, HCPs and bed managers were asked to rank barriers in relation to their perceived influence of the barriers in preventing the performance of the clinical care elements (influence attribute), and perceived difficulty in overcoming the barriers (difficulty attribute) [86]. In another study on the implementation of guideline recommendations for polypharmacy in multimorbid patients, barriers were prioritised based on 'relevance' and 'modifiability' [87]. When there is a large number of barriers present and participants are required to think of barriers against one or more criterion, it can be an overwhelming and exhausting process. In Aakhus, Oxman, & Flottorp's (2014) study, 352 determinants were identified and then prioritised based on ratings on the importance of the determinant (plausibility), and the extent to which the determinant could be addressed (feasibility) by the researchers [88]. While this approach was feasible to be conducted by the researchers themselves, it might be harder to execute with participants if there is a limited time to carry out the prioritisation exercise. More importantly, participants might feel fatigue when there is a large number of determinants involved resulting in the possibility of identifying inaccurate key barriers [89].

When developing implementation intervention, it is important to consider the amount of time of the development process will take as there is a possibility that barriers or implementation context may change over time. This was evident when the strategy of involving diabetes nurses to execute PDA delivery to patients was unable to be conducted at the point of implementation as there was a staff reorganisation in the hospital due to the lack of manpower. Therefore, the methods selected for identification and prioritisation of barriers should not take much time. The multivoting technique provides a quick and simple way to prioritise barriers and is a novel approach for development of tailored intervention. This step makes an important methodological contribution to intervention development.

## Engagement of stakeholders in mapping of strategies to barriers

In this study, stakeholders were involved in various steps of the intervention development including identification of barriers, prioritisation of barriers, and finalisation of the implementation intervention. However, they were not included in the initial process of identifying and

mapping of strategies to the prioritised barriers unlike in some other studies [78, 84, 90, 91]. Involving stakeholders in mapping strategies to barriers have been reported to be challenging in a study to identify effective strategies to integrate methadone maintenance therapy and HIV service in Vietnam. The difference in familiarity of context and strategies between the Vietnamese stakeholders and the implementation science experts affected scoring of the strategies as the stakeholders were unfamiliar with the strategies while the implementation science experts expressed that they lacked context-specific knowledge [84]. When selecting strategies to address barriers, it is important to understand the knowledge of both the organisational and cultural context as well as knowledge of the strategies so that their effectiveness is understood. Some studies addressed clinic stakeholders' lack of awareness and familiarity with strategies by presenting strategies identified in the literature to them and letting them select what they favour, or as a stimulus to create new strategies that would be more context-appropriate [78, 87, 91]. However, as alluded above, there may be a risk of information overload during the process that may render ineffective strategies mapped to barriers. Although the identification and mapping of strategies to barriers in this study was conducted by the researcher without involving the clinic stakeholders, one study has shown that the types of strategies generated by different stakeholders (healthcare researchers, healthcare professionals, quality improvement officers, healthcare authorities such as insurers or healthcare purchasers) for implementing evidence-based practice were similar [91]. Coupled with the literature review conducted on the effectiveness of strategies as well as informal observations and conversations with clinic staff, the researcher was well equipped with knowledge of strategies and the clinic context that enabled the identification and mapping of strategies to barriers to be performed with considerable rigour.

Even though the clinic stakeholders in this current study were not involved in the initial process of selecting and mapping of the strategies to the prioritised barriers, nevertheless, their input on potential strategies that can be adopted (findings from Phase 1) was taken into consideration. Furthermore, the draft intervention that was developed was also discussed with the clinic stakeholders and then refined based on their suggestions during the clinic stakeholders meeting. Using the findings of proposed strategies by participants in Phase 1 for intervention development was a practical strength and an efficient approach as it took into considerations the strategies and processes that were likely to be feasible to be implemented in the setting as well as helped to reduce the cost in terms of the need to regularly engage clinic stakeholders in the strategy-barrier mapping process, which could be resource intensive [92].

### Potential unintended consequences of the identified strategies

There is a possibility that some of the strategies identified may have unintended consequences. This study adopted the strategy 'Provide feedback' to motivate HCPs to adopt the insulin PDA. The expected mechanism of action for this strategy is social influences where doctors would be motivated when they know their colleagues are also using the insulin PDA. However, there is a possibility that doctors may feel pressured when receiving feedback on their performance and get stressed if they have a poorer performance when compared to other colleagues.

Another strategy that may cause unintended consequences is "Place the insulin PDA booklets in doctors' consultation rooms". For patients who are already averse to insulin therapy, the sight of the insulin PDA may make them feel uncomfortable or forced to start insulin therapy.

### Potential sustainability and scalability considerations for the implementation intervention

The implementation intervention development process in this study considered future sustainability and scalability. During strategy selection and operationalisation, while having a

champion had been found to be an effective strategy for PDA implementation [75], this strategy was not selected for this study because the UMMC primary care clinic has a regular turnover of medical officers due to it being part of an academic hospital. Hence, a trained champion might leave the clinic which rendered this strategy unsustainable. In addition, the researchers also ensured that the right staff were chosen to carry out the strategies as well as the timing and dosage of strategies. For example, the strategy 'educational workshop' would be conducted twice a year (dose) following doctor turnover in the clinic to ensure new doctors are trained on using the insulin PDA. Furthermore, engagement of stakeholders throughout various steps in this study in ensuring that strategies that were chosen target the right barriers, that they were feasible to be implemented in the clinic context, and integrating the implementation process with the diabetes patient care pathway in the clinic enhances the sustainability and scalability of the implementation intervention.

Many strategies identified in this study align with implementation science and healthcare quality improvement methodologies. For example, implementation science and healthcare quality improvement emphasizes on the importance of continuous monitoring and feedback for effective implementation. The strategy to provide feedback identified in this study helps to provide ongoing assessment of the implementation process for doctors, and ensuring that they are aware of their adoption and fidelity to the implementation protocol. Having explicit mandates is also a core principle of implementation science and quality improvement methods to drive the adoption of evidence-based practice. Leadership support can help to facilitate change, overcoming resistance, and creating a supportive environment for new practices. The strategy "mandate change" in this study aims to influence clinic staff to believe that the insulin PDA implementation is a clinic priority hence they would want to be involved in the implementation.

## Potential implications of study findings for policy and practice

This study highlights the potential strategies that can be implemented to integrate the use of insulin PDA in routine clinical practice. Healthcare providers would require training and support to integrate the insulin PDA in their practice. Effective implementation of the insulin PDA through the strategies identified can facilitate doctors in discussing diabetes management with patients thereby improving patient-doctor communication. Policymakers need to address systemic barriers through targeted policy changes, infrastructure improvements, or resource allocations for effective implementation of decision support tools. The use of PDA needs to be included in healthcare policies to support shared decision making practices among healthcare providers. In addition, policies can also be adjusted to consider patient satisfaction and engagement through the use of PDA as important metrics when evaluating the success of provision of quality health care.

## Limitations

This study has a few limitations. In Phase 1, some participants may not know the concept of SDM and PDAs. To address this barrier, participants were shown a video demonstrating how the insulin PDA can be used in a consultation, which may have biased their views about how to implement the PDA. Secondly, social desirability bias may have been present with participants who wanted to support the use of the PDAs or those that wanted to provide a favourable view of the clinic. This was addressed by ensuring participants that the data would be kept confidential and their participation would not affect their work, career or health care provision (for patients).

In Phase 2, the development of implementation intervention in this study involved only a few patient stakeholders and this may have affected the representativeness of the findings; i.e.,

when HCPs outnumber patients, it renders clinicians more visibility and perhaps they are also more vocal in the design of the intervention [93]. Nevertheless, care was taken to ensure that every stakeholder had a chance to voice their opinions by posing questions related to their roles in the implementation. This study also did not involve clinic staff such as medical assistants, and appointment clerks as part of the intervention development process given their limited roles in the implementation.

## Conclusion

This study highlights main barriers to PDA implementation, and strategies that can be adopted for implementation. The steps for intervention development in this study can be compared with other intervention development methods to advance the field of implementation of evidence-based innovations.

## Supporting information

**S1 Table. Voting form.**
(DOCX)

**S2 Table. Strategies proposed by Phase 1 participants to facilitate the insulin PDA implementation.**
(DOCX)

**S3 Table. Justification for strategies selected based on evidence in the literature and the clinic context.**
(DOCX)

**S4 Table. Strategies operationalised to address the prioritised barriers based on the clinic context.**
(DOCX)

**S1 Fig. The draft implementation intervention.**
(DOCX)

## Acknowledgments

The authors wish to thank the clinics involved in this study for the facilities and assistance provided.

## Author Contributions

**Conceptualization:** Wen Ting Tong, Yew Kong Lee, Chirk Jenn Ng, Ping Yein Lee.

**Data curation:** Wen Ting Tong, Yew Kong Lee, Chirk Jenn Ng, Ping Yein Lee.

**Formal analysis:** Wen Ting Tong, Yew Kong Lee, Chirk Jenn Ng, Ping Yein Lee.

**Funding acquisition:** Wen Ting Tong, Yew Kong Lee, Chirk Jenn Ng.

**Investigation:** Wen Ting Tong, Yew Kong Lee, Chirk Jenn Ng, Ping Yein Lee.

**Methodology:** Wen Ting Tong, Yew Kong Lee, Chirk Jenn Ng, Ping Yein Lee.

**Project administration:** Wen Ting Tong, Yew Kong Lee.

**Resources:** Wen Ting Tong.

**Software:** Wen Ting Tong.

**Supervision:** Yew Kong Lee, Chirk Jenn Ng.

**Validation:** Wen Ting Tong, Yew Kong Lee, Chirk Jenn Ng, Ping Yein Lee.

**Visualization:** Wen Ting Tong, Yew Kong Lee, Chirk Jenn Ng.

**Writing – original draft:** Wen Ting Tong.

**Writing – review & editing:** Wen Ting Tong, Yew Kong Lee, Chirk Jenn Ng, Ping Yein Lee.

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
