## [Decision Letter · Decision Letter 0]

23 Feb 2024

PONE-D-24-00184Development of an implementation intervention to integrate the use of health innovations in routine practice: an insulin patient decision aid as an exemplarPLOS ONE

Dear Dr. Tong,

Thank you for submitting your manuscript to PLOS ONE. After careful consideration, we feel that it has merit but does not fully meet PLOS ONE’s publication criteria as it currently stands. Therefore, we invite you to submit a revised version of the manuscript that addresses the points raised during the review process.

We look forward to receiving your revised manuscript.

Kind regards,

Rahul Priyadarshi

Academic Editor

PLOS ONE

Journal Requirements:

“This study was supported by the University of Malaya Research Grant (UMRG) (RP041C-15HTM); and the University of Malaya Postgraduate Research Grant (PPP) (PG264-2016A).”

Reviewers' comments:

Reviewer's Responses to Questions

**Comments to the Author**

1. Is the manuscript technically sound, and do the data support the conclusions?

Reviewer #1: Yes

Reviewer #2: Yes

2. Has the statistical analysis been performed appropriately and rigorously? 

Reviewer #1: Yes

Reviewer #2: Yes

3. Have the authors made all data underlying the findings in their manuscript fully available?

Reviewer #1: Yes

Reviewer #2: Yes

4. Is the manuscript presented in an intelligible fashion and written in standard English?

Reviewer #1: Yes

Reviewer #2: Yes

5. Review Comments to the Author

**Reviewer #1:** 1.Could you provide more clarity on the specific evidence-based innovation being implemented in routine practice?

2.Can you elaborate on why the Action Cycle in the Knowledge to Action framework was chosen as the guiding framework for intervention development?

3.How were participants recruited for the qualitative interviews, and what were the inclusion criteria?

4.Can you provide more detail on the interview guide developed based on the Theoretical Domains Framework?

5.How were the barriers and facilitators categorized into HCP, patient, organizational, and innovation factors?

6.Could you explain the process of multivoting in more detail, including how priorities were determined and by whom?

7.Can you provide examples or case studies illustrating how each of the identified strategies can address the prioritized barriers in the implementation of the insulin PDA?

8.How were the selected strategies operationalized and integrated into the patient care pathway, and were there any challenges encountered during this process?

9.What feedback was received during the clinic stakeholders meeting, and how was the final implementation intervention refined based on this input?

10.Can you discuss potential sustainability and scalability considerations for the implementation intervention beyond the initial phase?

11.What are the anticipated mechanisms through which each strategy is expected to facilitate the implementation of the insulin PDA?

12.Can you discuss the potential implications of the study findings for policy and practice, including recommendations for healthcare organizations and policymakers?

13.How do you plan to evaluate the effectiveness and impact of the implementation intervention in real-world settings?

14.Could you provide more information on the demographics and characteristics of the participants in the qualitative interviews?

15.Are there any biases or limitations associated with the qualitative data collection process, and if so, how were they addressed?

**Reviewer #2:** How were the 11 strategies identified to address the prioritized barriers, and what criteria were used for their selection?

Can you discuss any potential unintended consequences or limitations associated with the implementation of the identified strategies?

Were there any notable differences in the barriers and facilitators identified among different stakeholder groups (e.g., healthcare policymakers, doctors, patients)?

Can you provide more information on the training workshop strategy, including its content, duration, and intended audience?

How were the roles and responsibilities defined for the various stakeholders involved in the implementation of the insulin PDA?

Can you elaborate on the process of placing the insulin PDA in the consultation room and its potential impact on patient-provider interactions?

Were there any specific challenges or considerations related to engaging patients in treatment discussions during the implementation process?

Can you discuss how the identified strategies align with existing evidence on implementation science and healthcare quality improvement methodologies?

6. PLOS authors have the option to publish the peer review history of their article (what does this mean?). If published, this will include your full peer review and any attached files.

Reviewer #1: No

Reviewer #2: No

---

## [Author Response · Author response to Decision Letter 0]

26 Mar 2024

The Response to Reviewer has been uploaded as an attachment.

---

## [Decision Letter · Decision Letter 1]

12 Aug 2024

PONE-D-24-00184R1Development of an implementation intervention to integrate the use of health innovations in routine practice: an insulin patient decision aid as an exemplarPLOS ONE

Dear Dr. Tong,

Thank you for submitting your manuscript to PLOS ONE. After careful consideration, we feel that it has merit but does not fully meet PLOS ONE’s publication criteria as it currently stands. Therefore, we invite you to submit a revised version of the manuscript that addresses the points raised during the review process.

Please submit your revised manuscript by Sep 26 2024 11:59PM.  If you will need more time than this to complete your revisions, please reply to this message or contact the journal office at plosone@plos.org. Please include the following items when submitting your revised manuscript:A rebuttal letter that responds to each point raised by the academic editor and reviewer(s). You should upload this letter as a separate file labeled 'Response to Reviewers'.A marked-up copy of your manuscript that highlights changes made to the original version. You should upload this as a separate file labeled 'Revised Manuscript with Track Changes'.An unmarked version of your revised paper without tracked changes. You should upload this as a separate file labeled 'Manuscript'.If applicable, we recommend that you deposit your laboratory protocols in protocols.io to enhance the reproducibility of your results. Protocols.io assigns your protocol its own identifier (DOI) so that it can be cited independently in the future. For instructions see: https://journals.plos.org/plosone/s/submission-guidelines#loc-laboratory-protocols. Additionally, PLOS ONE offers an option for publishing peer-reviewed Lab Protocol articles, which describe protocols hosted on protocols.io. Read more information on sharing protocols at https://plos.org/protocols?utm_medium=editorial-email&utm_source=authorletters&utm_campaign=protocols.

We look forward to receiving your revised manuscript.

Kind regards,

Chiranjivi Adhikari, MPH, MHEd., PhD Candidate

Academic Editor

PLOS ONE

Journal Requirements:

Additional Editor Comments

Dear Authors,

Greetings!

It's a nice piece of scientific work. I thank the researchers' team and all the reviewers, for their comments during the review previously and now, along with the previous handling editor, for their academic contributions. I found earlier comments have been addressed satisfactorily. I, along with the current reviewers, I have the following comments.

1. Since PDA is our implementation intervention, it is the main point to be considered. As mentioned about the PDA and its booklet, various reviews have been cited regarding development and application in page 4, its cost-effectiveness and clinical efficacy were not mentioned elsewhere. I suggest reviewing some literature and highlighting.

2. It would be better to develop a figure of theory of change, that may reflect and synthesize all the steps, including formative and implementation phases, major barriers and strategies developed, and findings/conclusion. This may help readers understand easily.

3. Since this is implementation research, It would be better to reflect this in the title, as also, our one of the reviewers has suggested.

Kindly,

AE

Reviewers' comments:

Reviewer's Responses to Questions

**Comments to the Author**

1. If the authors have adequately addressed your comments raised in a previous round of review and you feel that this manuscript is now acceptable for publication, you may indicate that here to bypass the “Comments to the Author” section, enter your conflict of interest statement in the “Confidential to Editor” section, and submit your "Accept" recommendation.

Reviewer #1: (No Response)

Reviewer #3: All comments have been addressed

Reviewer #4: (No Response)

2. Is the manuscript technically sound, and do the data support the conclusions?

Reviewer #1: No

Reviewer #3: Yes

Reviewer #4: Yes

3. Has the statistical analysis been performed appropriately and rigorously? 

Reviewer #1: No

Reviewer #3: Yes

Reviewer #4: Yes

4. Have the authors made all data underlying the findings in their manuscript fully available?

Reviewer #1: (No Response)

Reviewer #3: Yes

Reviewer #4: Yes

5. Is the manuscript presented in an intelligible fashion and written in standard English?

Reviewer #1: No

Reviewer #3: Yes

Reviewer #4: Yes

6. Review Comments to the Author

Reviewer #1: (No Response)

Reviewer #3: Thank you to the authors for addressing comments raised. I recommend publication of this article. I have no further comments.

Reviewer #4: The study seems really unique having scientifically rigorous methodology. The manuscript is also well-organized and detailed in terms of methods, procedure and steps. Appreciating all your hard work and dedication, here are few comments to be addressed;

Comment 01: The title of manuscript may require minor revision as the study mainly focused on identifying implementation strategies and developing Implementation Intervention as an innovative approach rather than having use of innovation and integrating it into routine practice.

Comment 02: The study included urban government teaching hospital and community clinics, as different setting there might be different facilitators and barriers, thus how your study justifies the outcome ‘intervention’ for different settings as a whole?

Comment 03: ‘Understanding clinic context’ seems very pivotal step/phase in this study, thus what major areas were considered while examining clinic context that supports for further generalization of study?

7. PLOS authors have the option to publish the peer review history of their article (what does this mean?). If published, this will include your full peer review and any attached files.

Reviewer #1: No

Reviewer #3: No

Reviewer #4: No

---

## [Author Response · Author response to Decision Letter 1]

21 Aug 2024

Please refer to the "Response to Reviewers 2' file

---

## [Editor Report · Decision Letter 2]

5 Sep 2024

Developing an implementation intervention, and identifying strategies for integrating health innovations in routine practice: a case study of the implementation of an insulin patient decision aid

PONE-D-24-00184R2

Dear Dr. Wen Ting Tong,

We’re pleased to inform you that your manuscript has been judged scientifically suitable for publication and will be formally accepted for publication once it meets all outstanding technical requirements.

Kind regards,

Chiranjivi Adhikari, MPH, MHEd., PhD Candidate

Academic Editor

PLOS ONE
---

## [Editor Report · Acceptance letter]

13 Sep 2024

PONE-D-24-00184R2 

PLOS ONE

Dear Dr. Tong, 

I'm pleased to inform you that your manuscript has been deemed suitable for publication in PLOS ONE. Congratulations! Your manuscript is now being handed over to our production team.

Kind regards, 

on behalf of

Mr. Chiranjivi Adhikari 

Academic Editor

PLOS ONE